# Neural representation of action sequences: how far can a simple snippet-matching model take us?

**Cheston Tan**
Institute for Infocomm Research
Singapore
cheston@mit.edu

**Jedediah M. Singer**
Boston Children's Hospital
Boston, MA 02115
jedediah.singer@childrens.harvard.edu

**Thomas Serre     David Sheinberg**
Brown University
Providence, RI 02912
{Thomas_Serre, David_Sheinberg}@brown.edu

**Tomaso A. Poggio**
MIT
Cambridge, MA 02139
tp@ai.mit.edu

## Abstract

The macaque Superior Temporal Sulcus (STS) is a brain area that receives and integrates inputs from both the ventral and dorsal visual processing streams (thought to specialize in form and motion processing respectively). For the processing of articulated actions, prior work has shown that even a small population of STS neurons contains sufficient information for the decoding of actor invariant to action, action invariant to actor, as well as the specific conjunction of actor and action. This paper addresses two questions. First, what are the invariance properties of individual neural representations (rather than the population representation) in STS? Second, what are the neural encoding mechanisms that can produce such individual neural representations from streams of pixel images? We find that a simple model, one that simply computes a linear weighted sum of ventral and dorsal responses to short action "snippets", produces surprisingly good fits to the neural data. Interestingly, even using inputs from a single stream, both actor-invariance and action-invariance can be accounted for, by having different linear weights.

## 1   Introduction

For humans and other primates, action recognition is an important ability that facilitates social interaction, as well as recognition of threats and intentions. For action recognition, in addition to the challenge of position and scale invariance (which are common to many forms of visual recognition), there are additional challenges. The action being performed needs to be recognized in a manner invariant to the actor performing it. Conversely, the actor also needs to be recognized in a manner invariant to the action being performed. Ultimately, however, both the particular action and actor also need to be "bound" together by the visual system, so that the specific conjunction of a particular actor performing a particular action is recognized and experienced as a coherent percept.

For the "what is where" vision problem, one common simplification of the primate visual system is that the ventral stream handles the "what" problem, while the dorsal stream handles the "where" problem [1]. Here, we investigate the analogous "who is doing what" problem. Prior work has found that brain cells in the macaque Superior Temporal Sulcus (STS) — a brain area that receives converging inputs from dorsal and ventral streams — play a major role in solving the problem. Even with a small population subset of only about 120 neurons, STS contains sufficient information for action and actor to be decoded independently of one another [2]. Moreover, the particular conjunction of actor and action (i.e. stimulus-specific information) can also be decoded. In other words,

STS neurons have been shown to have successfully tackled the three challenges of actor-invariance, action-invariance and actor-action binding.

What sort of neural computations are performed by the visual system to achieve this feat is still an unsolved question. Singer and Sheinberg [2] performed population decoding from a collection of single neurons. However, they did not investigate the computational mechanisms underlying the individual neuron representations. In addition, they utilized a decoding model (i.e. one that models the usage of the STS neural information by downstream neurons). An encoding model – one that models the transformation of pixel inputs into the STS neural representation — was not investigated.

Here, we further analyze the neural data of [2] to investigate the characteristics of the neural representation at the level of individual neurons, rather than at the population level. We find that instead of distinct clusters of actor-invariant and action-invariant neurons, the neurons cover a broad, continuous range of invariance.

To the best of our knowledge, there have not been any prior attempts to predict single-neuron responses at such a high level in the visual processing hierarchy. Furthermore, attempts at time-series prediction for visual processing are also rare. Therefore, as a baseline, we propose a very simple and biologically-plausible encoding model and explore how far this model can go in terms of reproducing the neural responses in the STS. Despite its simplicity, modeling STS neurons as a linear weighted sum of inputs over a short temporal window produces surprisingly good fits to the data.

## 2 Background: the Superior Temporal Sulcus

The macaque visual system is commonly described as being separated into the ventral ("what") and dorsal ("where") streams [1]. The Superior Temporal Sulcus (STS) is a high-level brain area that receive inputs from both streams [3, 4]. In particular, it receives inputs from the highest levels of the processing hierarchy of either stream — inferotemporal (IT) cortex for the ventral stream, and the Medial Superior Temporal (MST) cortex for the dorsal stream. Accordingly, neurons that are biased more towards either encoding form information or motion information have been found in the STS [5]. The upper bank of the STS has been found to contain neurons more selective for motion, with some invariance to form [6, 7]. Relative to the upper bank, neurons in the lower bank of the STS have been found to be more sensitive to form, with some "snapshot" neurons selective for static poses within action sequences [7]. Using functional MRI (fMRI), neurons in the lower bank were found to respond to point-light figures [8] performing biological actions [9], consistent with the idea that actions can be recognized from distinctive static poses [10]. However, there is no clear, quantitative evidence for a neat separation between motion-sensitive, form-invariant neurons in the upper bank and form-sensitive, motion-invariant neurons in the lower bank. STS neurons have been found to be selective for specific combinations of form and motion [3, 11]. Similarly, based on fMRI data, the STS responds to both point-light display and video displays, consistent with the idea that the STS integrates both form and motion [12].

## 3 Materials and methods

**Neural recordings.** The neural data used in this work has previously been published by Singer and Sheinberg [2]. We summarize the key points here, and refer the reader to [2] for details. Two male rhesus macaques (monkeys $G$ and $S$) were trained to perform an action recognition task, while neural activity from a total of 119 single neurons (59 and 60 from $G$ and $S$ respectively) was recorded during task performance. The mean firing rate (FR) over repeated stimulus presentations was calculated, and the mean FR over time is termed the response "waveform" (Fig. 3). The monkeys' heads were fixed, but their eyes were free to move (other than fixating at the start of each trial).

**Stimuli and task.** The stimuli consisted of 64 movie clips (8 humanoid computer-generated "actors" each performing 8 actions; see Fig. 1). A sample movie of one actor performing the 8 actions can be found at http://www.jneurosci.org/content/30/8/3133/suppl/DC1 (see Supplemental Movie 1 therein). The monkeys' task was to categorize the action in the displayed clip into two predetermined but arbitrary groups, pressing one of two buttons to indicate their decision. At the start of each trial, after the monkey maintained fixation for 450ms, a blank screen was shown for 500ms, and then one of the actors was displayed (subtending $6°$ of visual angle vertically). Regardless of action, the actor was first displayed motionless in an upright neutral pose for 300ms, then began

performing one of the 8 actions. Each clip ended back at the initial neutral pose after 1900ms of motion. A button-press response at any point by the monkey immediately ended the trial, and the screen was blanked. In this paper, we considered only the data corresponding to the actions (i.e. excluding the motionless neutral pose). Similar to [2], we assumed that all neurons had a response latency of 130ms.

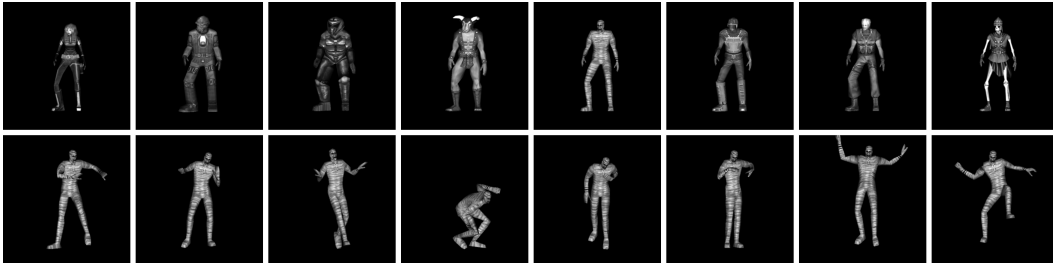

Figure 1: Depiction of stimuli used. Top row: the 8 "actors" in the neutral pose. Bottom row: sample frames of actor 5 performing the 8 actions; frames are from the same time-point within each action. The 64 stimuli were an 8-by-8 cross of each actor performing each action.

**Actor- and action-invariance indices.** We characterized each neuron's response characteristics along two dimensions: invariance to actor and to action. For the actor-invariance index, a neuron's average response waveform to each of the 8 actions was first calculated by averaging over all actors. Then, we calculated the Pearson correlation between the neuron's actual responses and the responses that would be seen if the neuron were completely actor-invariant (i.e. if it always responded with the average waveform calculated in the previous step). The action-invariance index was calculated similarly. The calculation of these indices bear similarities to that for the pattern and component indices of cells in area MT [13].

**Ventral and dorsal stream encoding models.** We utilize existing models of brain areas that provide input to the STS. Specifically, we use the HMAX family of models, which include models of the ventral [14] and dorsal [15] streams. These models receive pixel images as input, and simulate visual processing up to areas V4/IT (ventral) and areas MT/MST (dorsal). Such models build hierarchies of increasingly complex and invariant representations, similar to convolutional and deep-learning networks. While ventral stream processing has traditionally been modeled as producing outputs in response to static images, in practice, neurons in the ventral stream are also sensitive to temporal aspects [16]. As such, we extend the ventral stream model to be more biologically realistic. Specifically, the V1 neurons that project to the ventral stream now have temporal receptive fields (RFs) [17], not just spatial ones. These temporal and spatial RFs are separable, unlike those for V1 neurons that project to the dorsal stream [18]. Such space-time separable V1 neurons that project to the ventral stream are not directionally-selective and are not sensitive to motion per se. They are still sensitive to form rather than motion, but are better models of form processing, since in reality input to the visual system consists of a continuous stream of images. Importantly, the parameters of dorsal and ventral encoding models were fixed, and there was no optimization done to produce better fits to the current data. We used only the highest-level ($C2$) outputs of these models.

**STS encoding model.** As a first approximation, we model the neural processing by STS neurons as a linear weighted sum of inputs. The weights are fixed, and do not change over time. In other words, at any point in time, the output of a model STS neuron is a linear combination of the *C2* outputs produced by the ventral and dorsal encoding models. We do not take into account temporal phenomena such as adaptation. We make the simplifying (but unrealistic) assumptions that synaptic efficacy is constant (i.e. no "neural fatigue"), and time-points are all independent.

Each model neuron has its own set of static weights that determine its unique pattern of neural responses to the 64 action clips. The weights are learned using leave-one-out cross-validation. Of the 64 stimuli, we use 63 for training, and use the learnt weights to predict the neural response waveform to the left-out stimulus. This procedure is repeated 64 times, leaving out a different stimulus each time. The 64 sets of predicted waveforms are collectively compared to the original neural responses. The goodness-of-fit metric is the Pearson correlation ($r$) between predicted and actual responses.

The weights are learned using simple linear regression. For number of input features $F$, there are $F + 1$ unknown weights (including a constant bias term). The inputs to the STS model neuron are represented as a $(T \times 63)$ by $(F + 1)$ matrix, where $T$ is the number of timesteps. The output is a $(T \times 63)$ by 1 vector, which is simply a concatenation of the 63 neural response waveforms corresponding to the 63 training stimuli. This simple linear system of equations, with $(F + 1)$ unknowns and $(T \times 63)$ equations, can be solved using various methods. In practice, we used the least-squares method. Importantly, at no point are ground-truth actor or action labels used.

Rather than use the 960 dorsal and/or 960 ventral *C2* features directly as inputs to the linear regression, we first performed PCA on these features (separately for the two streams) to reduce the dimensionality. Only the first 300 principal components (accounting for 95% or more of the variance) were used; the rest was discarded. Therefore, $F = 300$. Fitting was also performed using the combination of dorsal and ventral *C2* features. As before, PCA was performed, and only the first 300 principal components were retained. Keeping $F$ constant at 300, rather than setting it to 600, allowed for a fairer comparison to using either stream alone.

## 4   What is the individual neural representation like?

In this section, we examine the neural representation at the level of individual neurons. Figure 2 shows the invariance characteristics of the population of 119 neurons. Overall, the neurons span a broad range of action and actor invariance (95% of invariance index values span the ranges [0.301 0.873] and [0.396 0.894] respectively). The correlation between the two indices is low ($r$=0.26). Considering each monkey separately, the correlations between the two indices were 0.55 (monkey $G$) and -0.09 (monkey $S$). This difference could be linked to slightly different recording regions [2].

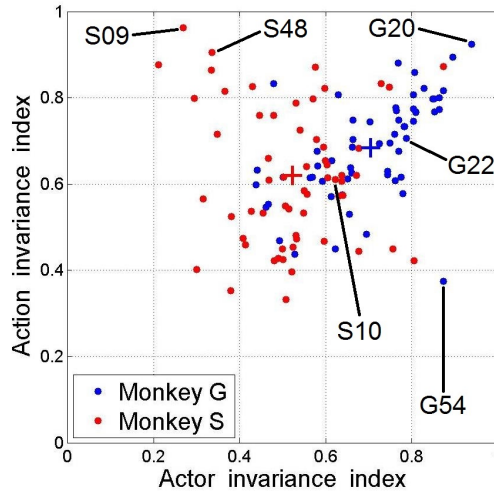

Figure 2: Actor- and action-invariance indices for 59 neurons from monkey $G$ (blue) and 60 neurons from monkey $S$ (red). Blue and red crosses indicate mean values.

Figure 3 shows the response waveforms of some example neurons to give a sense of what response patterns correspond to low and high invariance indices. The average over actors, average over actions and the overall average are also shown. Neuron $S09$ is highly action-invariant but not actor-invariant, while $G54$ is the opposite. Neuron $G20$ is highly invariant to both action and actor, while the invariance of $S10$ is close to the mean invariance of the population.

We find that there are no distinct clusters of neurons with high actor-invariance or action-invariance. Such clusters would correspond to a representation scheme in which certain neurons specialize in coding for action invariant to actor, and vice-versa. A cluster of neurons with both low actor- and action-invariance could correspond to cells that code for a specific conjunction (binding) of actor and action, but no such cluster is seen. Rather, Fig. 2 indicates that instead of the "cell specialization" approach to neural representation, the visual system adopts a more continuous and distributed representation scheme, one that is perhaps more universal and generalizes better to novel stimuli. In the

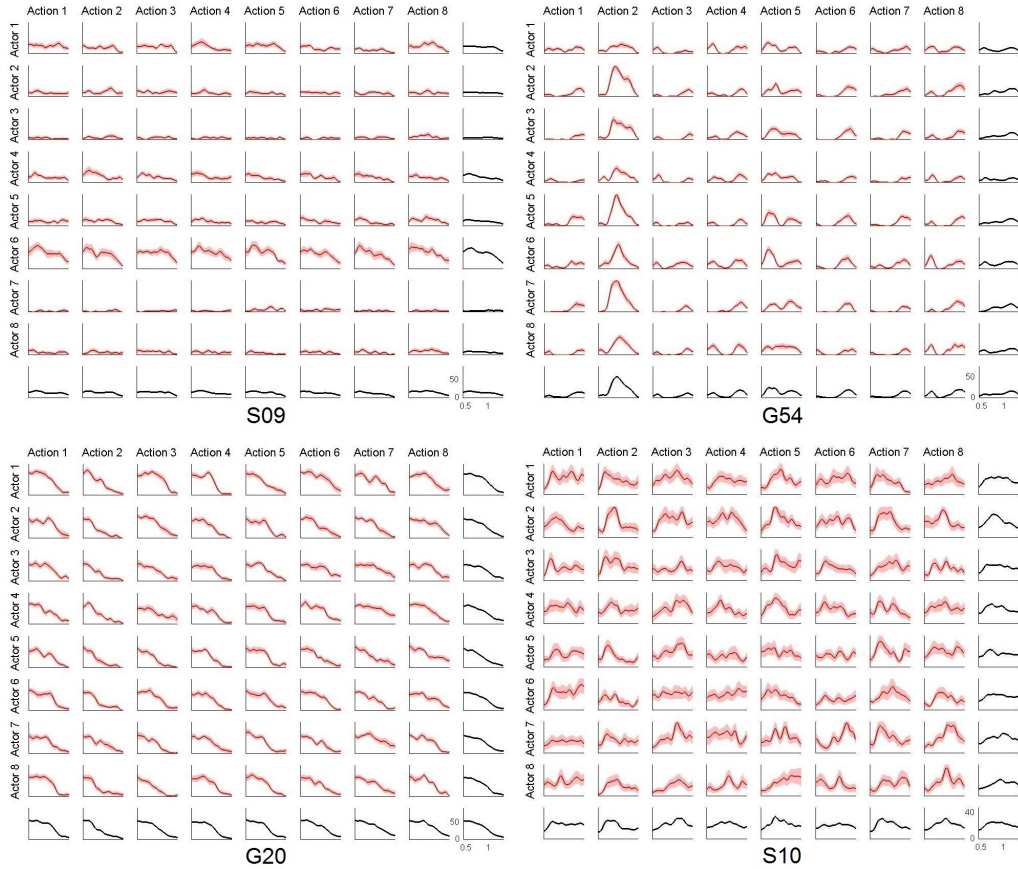

Figure 3: Plots of waveforms (mean firing rate in Hz vs. time in secs) for four example neurons. Rows are actors, columns are actions. Red lines: mean firing rate (FR). Light red shading: ±1 SEM of FR. Black lines (row 9 and column 9): waveforms averaged over actors, actions, or both.

rest of this paper, we explore how well a linear, feedforward encoding model of STS ventral/dorsal integration can reproduce the neural responses and invariance properties found here.

# 5 The "snippet-matching" model

In their paper, Singer and Sheinberg found evidence for the neural population representing actions as "sequences of integrated poses" [2]. Each pose contains visual information integrated over a window of about 120ms. However, it was unclear what the representation was for each individual neuron. For example, does each neuron encode just a single pose (i.e. a "snippet"), or can it encode more than one? What are the neural computations underlying this encoding?

In this paper, we examine what is probably the simplest model of such neural computations, which we call the "snippet-matching" model. According to this model, each individual STS neuron compares its incoming input over a single time step to its preferred stimulus. Due to hierarchical organization, this single time step at the STS level contains information processed from roughly 120ms of raw visual input. For example, a neuron matches the incoming visual input to one particular short segment of the human walking gait cycle, and its output at any time is in effect how similar the visual input (from the previous 120ms up to the current time) is compared to that preferred stimulus (represented by linear weights; see sub-section on STS encoding model in Section 3).

Such a model is purely feedforward and does not rely on any lateral or recurrent neural connections. The temporal-window matching is straightforward to implement neurally e.g. using the same "delay-line" mechanisms [19] proposed for motion-selective cells in V1 and MT. Neurons implementing

this model are said to be "memoryless" or "stateless", because their outputs are solely dependent on their current inputs, and not on their own previous outputs. It is important to note that the inputs to be matched could, in theory, be very short. In the extreme, the temporal window is very small, and the visual input to be matched could simply be the current frame. In this extreme case, action recognition is performed by the matching of individual frames to the neuron's preferred stimulus.

Such a "snippet" framework (memoryless matching over a single time step) is consistent with prior findings regarding recognition of biological motion. For instance, it has been found that humans can easily recognize videos of people walking from short clips of point-light stimuli [8]. This is consistent with the idea that action recognition is performed via matching of snippets. Neurons sensitive to such action snippets have been found using techniques such as fMRI [9, 12] and electrophysiology [7]. However, such snippet encoding models have not been investigated in much detail.

While there is some evidence for the snippet model in terms of the existence of neurons responsive and selective for short action sequences, it is still unclear how feasible such an encoding model is. For instance, given some visual input, if a neuron simply tries to match that sequence to its preferred stimulus, how exactly does the neuron ignore the motion aspects (to recognize actor invariant to action) or ignore the form aspects (to recognize action invariant to actors)? Given the broad range of actor- and action-invariances found in the previous section, it is crucial to see if the snippet model can in fact reproduce such characteristics.

## 6 How far can snippet-matching go?

In this section, we explore how well the simple snippet-matching model can predict the response waveforms of our population of STS neurons. This is a challenging task. STS is high up in the visual processing hierarchy, meaning that there are more unknown processing steps and parameters between the retina and STS, as compared to a lower-level visual area. Furthermore, there is a diversity of neural response patterns, both between different neurons (see Figs. 2 and 3) and sometimes also between different stimuli for a neuron (e.g. $S10$, Fig. 3).

The snippet-matching process can utilize a variety of matching functions. Again, we try the simplest possible function: a linear weighted sum. First, we examine the results of the leave-one-out fitting procedure when the inputs to STS model neurons are from either the dorsal or ventral streams alone. For monkey $G$, the mean goodness-of-fit (correlation between actual and predicted neural responses on left-out test stimuli) over all 59 neurons are 0.50 and 0.43 for the dorsal and ventral stream inputs respectively. The goodness-of-fit is highly correlated between the two streams ($r$=0.94). For monkey $S$, the mean goodness-of-fit over all 60 neurons is 0.33 for either stream (correlation between streams, $r$=0.91). Averaged over all neurons and both streams, the mean goodness-of-fit is 0.40. As a sanity check, when either the linear weights or the predictions are randomly re-ordered, mean goodness-of-fit is 0.00.

Figure 4 shows the predicted and actual responses for two example neurons, one with a relatively good fit ($G22$ fit to dorsal, $r$=0.70) and one with an average fit ($S10$ fit to dorsal, $r$=0.39). In the case of $G22$ (Fig. 4 left), which is not even the best-fit neuron, there is a surprisingly good fit despite the clear complexity in the neural responses. This complexity is seen most clearly from the responses to the 8 actions averaged over actors, where the number and height of peaks in the waveform vary considerably from one action to another. The fit is remarkable considering the simplicity of the snippet model, in which there is only one set of static linear weights; all fluctuations in the predicted waveforms arise purely from changes in the inputs to this model STS neuron.

Over the whole population, the fits to the dorsal model (mean $r$=0.42) are better than to the ventral model (mean $r$=0.38). Is there a systematic relationship between the difference in goodness-of-fit to the two streams and the invariance indices calculated in Section 4? For instance, one might expect that neurons with high actor-invariance would be better fit to the dorsal than ventral model. From Fig. 5, we see that this is exactly the case for actor invariance. There is a strong positive correlation between actor invariance and difference (dorsal minus ventral) in goodness-of-fit (monkey $G$: $r$=0.72; monkey $S$: $r$=0.69). For action invariance, as expected, there is a negative correlation (i.e. strong action invariance predicts better fit to ventral model) for monkey $S$ ($r$=-0.35). However, for monkey $G$, the correlation is moderately positive ($r$=0.43), contrary to expectation. It is unclear why this is the case, but it may be linked to the robust correlation between actor- and action-invariance indices for monkey $G$ ($r$=0.55), seen in Fig. 2. This is not the case for monkey $S$ ($r$=-0.09).

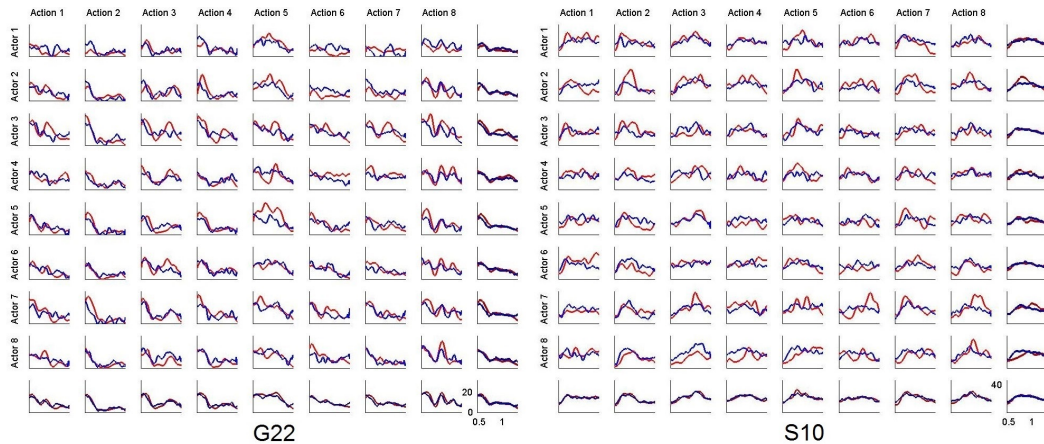

Figure 4: Predicted (blue) and actual (red) waveforms for two example neurons, both fit to the dorsal stream. $G22$: $r$=0.70, $S10$: $r$=0.39. For each of the 64 sub-plots, the prediction for that test stimulus used the other 63 stimuli for training. Solely for visualization purposes, predictions were smoothed using a moving average window of 4 timesteps (total length 89 timesteps).

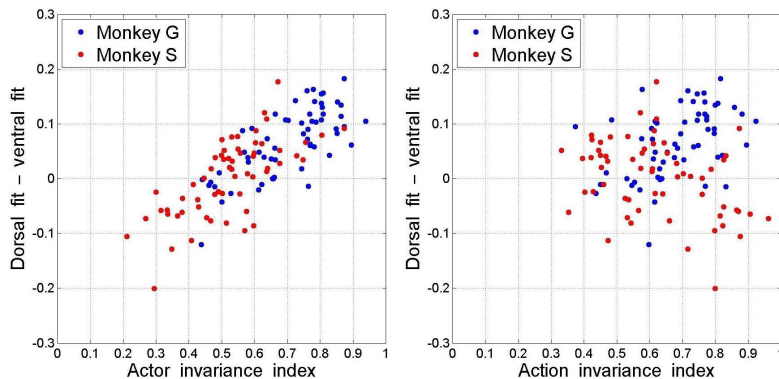

Figure 5: Relationship between goodness-of-fit and invariance. Y-axis: difference between $r$ from fitting to dorsal versus ventral streams. X-axis: actor (left) and action (right) invariance indices.

Interestingly, either stream can produce actor-invariant and action-invariant responses (Fig. 6). While $G54$ is better fit to the dorsal than ventral stream (0.77 vs. 0.67), both fits are relatively good — and are actor-invariant. The converse is true for $S48$. These results are consistent with the reality that both streams are interconnected and the what/where distinction is a simplification.

So far, we have performed linear fitting using the dorsal and ventral streams separately. Does fitting to a combination of both models improve the fit? For monkey $G$, the mean goodness-of-fit is 0.53; for monkey $S$ it is 0.38. The improvements over the better of either model alone are moderate (6% for $G$, 15% for $S$). Interestingly, this fitting to a combination of streams without prior knowledge of which stream is more suitable, produces fits that are as good or better than if we knew a priori which stream would produce a better fit for a specific neuron (0.53 vs. 0.51 for $G$; 0.38 vs. 0.36 for $S$).

How much better compared to low-level controls does our snippet model fit to the combined outputs of dorsal and ventral stream models? To answer this question, we instead fit our snippet model to a low-level pixel representation while keeping all else constant. The stimuli were resized to be 32 x 32 pixels, so that the number of features (1024 = 32 x 32) was roughly the same number as the $C2$ features. This was then reduced to 300 principal components, as was done for $C2$ features. Fitting our snippet model to this pixel-derived representation produced worse fits ($G$: 0.40, $S$: 0.32). These were 25% ($G$) and 16% ($S$) worse than fitting to the combination of dorsal and ventral models. Furthermore, the monkeys were free to move their eyes during the task (apart from a fixation period

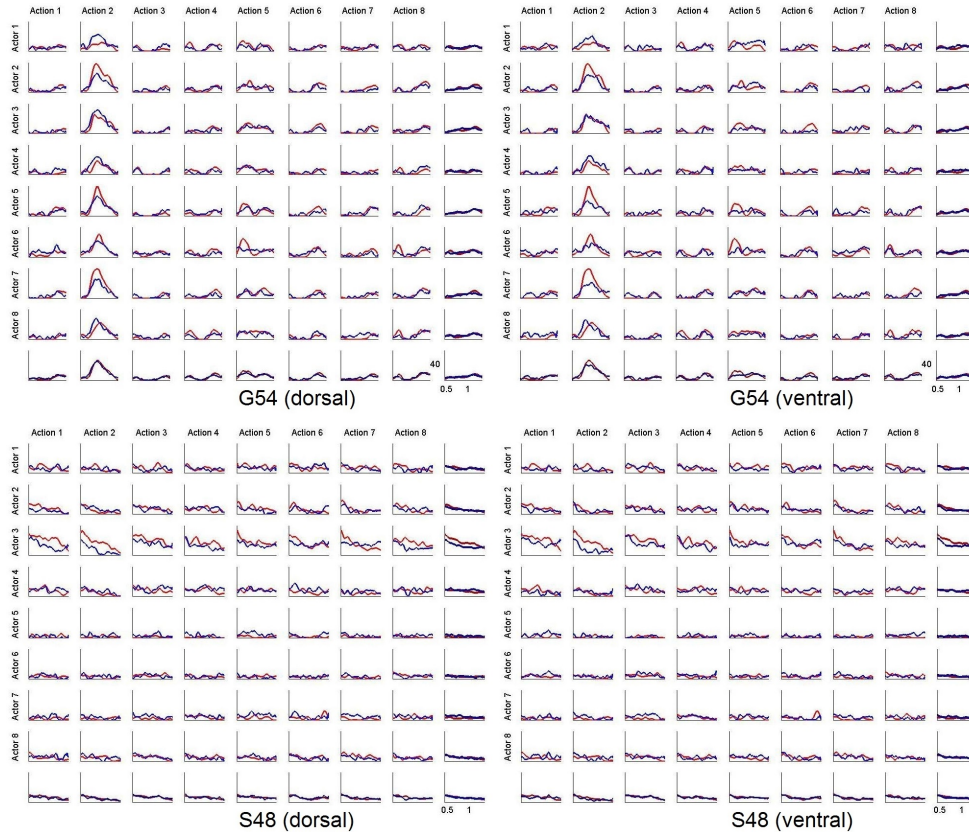

Figure 6: Either stream can produce actor-invariant ($G54$) and action-invariant ($S48$) responses.

at the start of each trial). Even slight random shifts in the pixel-derived representation of less than $0.25°$ of visual angle (on the order of micro-saccades) dramatically reduced the fits to 0.25 ($G$) and 0.21 ($S$). In contrast, the same random shifts did not change the average fit numbers for the combination of dorsal and ventral models (0.53 for $G$, 0.39 for $S$). These results suggest that the fitting process does in fact learn meaningful weights, and that biologically-realistic, robust encoding models are important in providing suitable inputs to the fitting process.

Finally, how do the actor- and action-invariance indices calculated from the predicted responses compare to those calculated from the ground-truth data? Averaged over all 119 neurons fitted to a combination of dorsal and ventral streams, the actor- and action-invariance indices are within 0.0524 and 0.0542 of their true values (mean absolute error). In contrast, using the pixel-derived representation, the results are much worse (0.0944 and 0.1193 respectively, i.e. the error is double).

## 7 Conclusions

We found that at the level of individual neurons, the neuronal representation in STS spans a broad, continuous range of actor- and action-invariance, rather than having groups of neurons with distinct invariance properties. Simply as a baseline model, we investigated how well a linear weighted sum of dorsal and ventral stream responses to action "snippets" could reproduce the neural response patterns found in these STS neurons. The results are surprisingly good for such a simple model, consistent with findings from computer vision [20]. Clearly, however, more complex models should, in theory, be able to better fit the data. For example, a non-linear operation can be added, as in the LN family of models [13]. Other models include those with nonlinear dynamics, as well as lateral and feedback connections [21, 22]. Other ventral and dorsal models can also be tested (e.g. [23]), including computer vision models [24, 25]. Nonetheless, this simple "snippet-matching" model is able to grossly reproduce the pattern of neural responses and invariance properties found in the STS.

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
