[Reviews · NeurIPS 2013]

Submitted by Assigned_Reviewer_5

Paper 362 "Neural representation of action sequences: how far can a simple snippet-matching model take us?"
The paper describes how a relatively simple model of "what" and "where" processing into "who is doing what" stream can already account for a sizeable amount of the neural responses in area STS.

Quality. The paper is clear and concise, with the right amount of detail to understand the model, building on well established data and models.

Clarity. The paper is well written and easy to follow. A minor issue is that on paper, the waveform plots are taxing on the eyes. One other minor issue is that I failed to exactly understand what "snippets" are: is this just a temporal filtering operation with a finite length filter, or is this about filtering just pre-segmented non-overlapping motion windows? Clarification would be appreciated.

Originality. To the best of my knowledge, the paper presents the first reasonably successful model of neural responses in STS.
Significance. This paper fits into a stream of recent work aiming at modeling and understanding the deeper areas of neural information processing, in this case for area STS. Our understanding of these areas is very limited, both of the exact type of computation that is carried out, and how a succession of neural areas arrives at such computations. This paper significantly contributes to understanding how ventral and dorsals streams may be integrated, allowing for testable predictions as well a potential basis for deep vision applications. A minor quibble is that it would have been interesting to see how the STS model performs for action/actor recognition tasks.
Summary: This paper describes a deep model for integrating "what" and "where" processing in the brain, and demonstrates that such a model fits STS neural responses already to a remarkable degree.

Submitted by Assigned_Reviewer_6

This paper focusses on addressing the problem of "who is doing what", i.e
how action sequences are processed. The authors focus their attention on
the superior temporal sulcus (STS) as this particular brain area has been
previously implicated in playing a major role in solving the problem.
Specifically, they propose a simple neural encoding based on simple linear
weightings, and show that it is capable of producing good fits to the
neurophysiological data recorded from macaques engaged in a action
categorization task.

This work is nice extension of some previous work by Sheinberg and Singer,
the primary difference being the use of a neural encoding rather than
decoding model. In some sense, it seems like occam's razor is at play, in that
the simple model may be the best. It does seem puzzling to me that such a
simplistic linear model can perform so well at such a high level in the
visual pathway, especially considering linear encoding models do not typically fair
well in V1 (for example). In my opinion, this paper seems more like a
really good starting off point for a far more interesting project - it
seems clear that there are a lot of questions still to be answered,
primarily through the use of more complex models that can more accurately
fit the data, and be more biologically plausible.

The paper itself is written in a relatively clear manner. That said, one
thing that did frustrate me to some degree was the lack of information
given about the HMAX family of models. This family of models is crucial to
the understanding of the modelling approach used within the paper, so it
would have been nice to see them explained in somewhat more detail.
Summary: This is an interesting paper looking at neural encoding in a higher visual area using a simplistic linear model. Although the results are somewhat compelling, it is clear that there is a lot more work to be done.

Submitted by Assigned_Reviewer_7

SUMMARY

This paper makes two points. First individual neurons in superior temporal sulcus have a continuous and distributed encoding of both the actor and action (and their conjunction) during action observation. It then makes the second point that the evoked responses (mean firing rate) can be predicted by a linear mixture of transformed pixel input – where the transform is motivated by existing models of dorsal and ventral stream processing. The report is written extremely nicely; however, it is rather descriptive and could be improved with some clarification of its conceptual direction and analysis details.


COMMENTS TO AUTHORS

Your report was written very nicely and enjoyable to read. On the other hand, there was a slight ambiguity about the underlying message and the connection between the snippet-matching model and your analyses. Perhaps you could consider the following;

1) Your analyses make two separate points. First, there are action and actor invariant responses in STS neurons. Second, you can predict the responses of STS neurons using a transformation of pixel inputs which – at the last stage – involves a linear mixture. My problem here is that these two points are unconnected. You could have presented the invariance analyses separately from the linear mixture analyses and vice versa. This is not necessarily bad but I think you need to clarify what the contribution of this report is and try to link the two points through some contribution to our understanding of perceptual synthesis?

2) It is not clear how your linear regression analysis relates to the snippet-matching model. Your discussion of the snippet-matching model could be interpreted as either a response to a short sequence of inputs (with explicit temporal support) or simply a response to a time average (integrating the input over a small time window). Given that you seem to interpret your regression analysis in reference to the snippet-matching model, I suspect the latter is the case. If this is so, could you make it clear that each time bin or time step in the regression analysis corresponds to a snippet of input and that this is summarized by the average over 120 milliseconds. Furthermore, say how many time steps there were – in other words quantify T. It is confusing because you introduce snippet-matching after the regression analysis and then say: “Again, we try the simplest possibly function:” When you say “Again”, does this mean you have done two analyses and have only reported one or are you simply interpreting regression analysis in terms of the snippet-matching model. If the latter, it might be useful to introduce the snippet-matching model first and then describe how it motivated your linear regression.

MINOR POINTS

1) In the abstract, you need to make it clear that this paper is about characterising neuronal responses to action observation. The title and first sentences made me think I was going to read about motor control.

2) I am not sure about your argument in the second paragraph of the introduction. Simply showing that there is information about actions, actors (and their conjunction) in any part of the brain does not mean they have tackled a “(three-fold) challenge”. One would find (presumably) very similar response properties in any part of the mirror neuron system. Furthermore, if one analysed retinal cells, one would also find this information. Perhaps you could highlight the fact that you have found information or invariance properties at the level of the single neuron - that could not be found at low levels in the visual hierarchy - to make your point more clearly?

3) Your use of cross validation to establish the utility of the linear regression model of neuronal responses is fine for the machine learning community. However, it is statistically sub-optimal in relation to inference based directly on your general linear model: by central limit theorem, you can assume Gaussian errors and provide a much more sensitive analysis in terms of classical inference. Furthermore, you could have used conventional F statistics to test different hypotheses about the relative contribution of dorsal and ventral predictors of the neuronal responses – or indeed their interaction. I suspect you will choose to stay with the cross validation scheme but it might be interesting to pursue standard statistical analyses in future work?

4) On page 5 (line 250) I would say “A more continuous and distributed representation scheme”.

5) On page 6 (line 299) what is “a matching function”? The notion of a snippet-matching process is not described very clearly. Is this simply an instantaneous mapping between some running average? Or is there something deeper going on?

6) Page 5 (line 304), I would spell out that the dorsal and ventral predictors both had a correlation coefficient of 0.33 – otherwise, people will think this is the correlation coefficient for the combined analysis.

7) You do not report any statistical inference or p-values. It might be nice to associate p-values with your correlation coefficients and use a Fishers transform when comparing the correlations between different regression models (for example, the processed versus non-processed principal components).

I hope that these comments help should any revision be required.

Summary: This was an interesting but descriptive characterisation of STS neuronal responses to action observation. It highlights some simple invariance and response properties using a descriptive linear analysis (where predicator
Author Feedback

Author rebuttal: We thank the reviewers for their thoughtful comments. We find it encouraging that all reviewers agree that the paper is of sufficient quality for NIPS. Thus, we will briefly reiterate the main contributions and novelty here, and put responses to specific comments after the main text.

One key contribution, as pointed out by the first reviewer, is that "the paper presents the first reasonably successful model of neural responses in STS", a brain area that is high up the hierarchy of visual areas. Such higher areas and their neural computations are poorly understood. Our paper demonstrates empirically that a linear combination of the outputs of preceding neural layers can go surprisingly far.

Our paper also demonstrates the interesting points that different linear combinations of the same "basis waveforms" can produce either actor-invariance or action-invariance, and that interpreting the "integration" of dorsal and ventral streams as a linear weighted sum may be a reasonable first-order approximation.

These results tie-in nicely with other hierarchical models whose elementary operations are relatively simple, yet when composed together hierarchically can perform complex computations (e.g. deep learning methods).

Furthermore, the paper is one of the few instances of time-series prediction, and one of the few quantitative characterizations of single-neuron invariance to properties other than position and scale.

In sum, we feel that this paper breaks new ground on several fronts, and also presents several interesting empirical findings that are still not well-understood (the explanation of which may motivate much further work).

Thank you.

====

FIRST REVIEWER

COMMENT: Waveforms are taxing
RESPONSE: We are open to suggestions. The plots are important for giving a sense of what the invariance indices and r values really mean. We tried to optimize the viewing experience through a careful choice of color, size, placement, etc.

COMMENT: What exactly are snippets?
RESPONSE: Generally, a snippet is a vague concept of a short temporal window. Here, it is simply one time step's worth of output from the previous layer. This, in theory, could contain information from an arbitrarily long temporal window (depending on the "integration time" of a neuron, and because of the hierarchical structure). Practically speaking, given the (prior, fixed) parameters of the HMAX models, a snippet corresponds to roughly 120ms of raw input video.

COMMENT: Performance on actor/action recognition
RESPONSE: This would be interesting. Space permitting (we are maxed out), we could report a few numbers, but a proper treatment would best be left to a future paper.


====

SECOND REVIEWER

COMMENT: Puzzling that simple linear model performs so well
RESPONSE: While we don't claim to fully understand the surprising level of performance, we note that:
1) hierarchical models composed from simple operations (e.g. Rust et al., Nature Neurosci 2006) have also achieved reasonable-to-good performance in other brain areas e.g. MT
2) most prior papers in V1 do something slightly different: train on a subset of repetitions, test on left-out repetitions. There, performance is severely limited by noise. Here, we use firing rates averaged over *all repetitions*, and leave out *stimuli*. Noise is less of a limiting factor. We have also done the "leave repetitions out" analysis (not reported due to space), and r is about 0.35 (a bit less than typical V1 fits).

COMMENT: Lack of details about HMAX
RESPONSE: Due to lack of space, we omitted HMAX details, which have been reported elsewhere (see references). Also, we do not focus on HMAX in this paper, and other hierarchical models may achieve similar results. But we will try to add more details if space permits.

COMMENT: "Paper seems more like a really good starting off point for a far more interesting project."
RESPONSE: We somewhat agree, and note that sometimes the best papers provide more questions than answers (e.g. by reporting puzzling empirical results, which generate impact through attempts at explanation). We also note that in modeling, sometimes extra bells and whistles only provide marginally better results.


====

THIRD REVIEWER

COMMENT: "interesting but descriptive"
RESPONSE: We agree, and note that in neuroscience, descriptive work is often an important and necessary first step to establish certain empirical findings, which then motivates further work. In this paper, we have made several interesting findings, which we have yet to understand fully, and which we hope follow-up work can explain.

COMMENT: "You could have presented the invariance analyses separately from the linear mixture analyses... try to link the two points..."
RESPONSE: The two analyses are indeed somewhat separate. The linkage (and we will try to make this clearer) is that the linear-mixture model can reproduce the waveforms relatively well (and hence also the invariance indices; line 413), despite the diversity in invariance properties (both types of highly-invariant neurons are generally well-fitted).

COMMENT: Link between linear regression and snippet-matching
RESPONSE: We apologize for the disjoint between descriptions of linear regression and snippet-matching. We will edit to clarify. Snippet-matching relates to the size of the temporal window for matching actual to preferred inputs. In this paper, the specific metric for matching is a linear weighted sum, i.e. linear regression can be used to recover weights. Yes, "each time bin or time step in the regression analysis corresponds to a snippet of input", but the input to an STS neuron during one time step is the result of hierarchically processing roughly 120ms of raw input.

RESPONSE TO MINOR POINTS: We appreciate the comments, and will try to (space permitting):
- clarify about motor control, the three-fold challenge, invariance, etc.
- add statistical analyses in future work